# Persistent Reverse End Diastolic Flow in Fetal Middle Cerebral Artery: A Rare Finding with Poor Outcomes

**DOI:** 10.3390/medicina59091648

**Published:** 2023-09-12

**Authors:** Dani Setiawan, Johanes Cornelius Mose, Nuniek Kharismawati, Fatima Zahra, Muhammad Gilang Dwi Putra

**Affiliations:** 1Maternal Fetal Medicine Division, Department of Obstetrics and Gynecology, Faculty of Medicine, Hasan Sadikin Central General Hospital, Padjadjaran University, Bandung 40161, Indonesia; nukniek@gmail.com (N.K.); kucingtaksis@gmail.com (F.Z.); 2Child Health Department, Faculty of Medicine, Hasan Sadikin Central General Hospital, Padjadjaran University, Bandung 40161, Indonesia; gilangdwiputra@gmail.com

**Keywords:** persistent reversed end diastolic flow, middle cerebral artery

## Abstract

Doppler findings of persistent reverse end-diastolic flow (PREDF) in a fetal middle cerebral artery (MCA) are a very rare sonographic finding and are a marker of poor fetal condition. This finding often leads to intrauterine fetal death or early neonatal death. Reverse end-diastolic flow in the middle cerebral artery is an advanced hemodynamic event. Fetal cerebral circulation normally has a high impedance; in the event of fetal hypoxemia, impedance decreases, resulting in the central redistribution of blood flow to vital organs, which maintains the oxygen delivery to the brain. Reverse flow in the middle cerebral arteries describes the loss of this autoregulatory process. PREDF is a sequence that occurs due to increased extracranial or intracranial pressure. Previous case reports mentioned intracranial hemorrhage, fetal growth restriction, fetal anemia, and fetal hepatic abnormalities as problems leading to PREDF. This condition presumably arises due to cerebral edema associated with severe hypoxemia. We reported Doppler findings of PREDF MCA in a 33-year-old female patient at 30 weeks gestation who was referred to the hospital with severe preeclampsia accompanied by fetal growth restriction and oligohydramnios. A cesarean section was performed due to severe preeclampsia and a low bishop score. Hypotheses on various etiologies and their association with intrauterine/neonatal death as well as the best management still require further investigation.

## 1. Introduction

Middle cerebral arteries (MCAs) are the most accessible cerebral vessel in fetal Doppler examination because they carry 80% of cerebral blood flow. The right and left cerebral arteries are the main branches of the Circulus Willis that originate from the carotid and vertebral arteries [1]. Fetal cerebral circulation normally has a high impedance; in the event of fetal hypoxemia, impedance decreases, resulting in the central redistribution of blood flow to vital organs, which maintains the oxygen delivery to the brain. A decrease in impedance manifests with a decrease in cerebral arterial resistance, which is an early marker of autoregulation in hypoxemic states [2]. This redistribution is known as the brain-sparing effect and plays an important role in the adaptation of the fetus to the lack of oxygen [1].

Reverse end-diastolic flow in the MCA is a rare fetal condition and an advanced hemodynamic event [3]. PREDF of the MCA occurs as a result of increased intracranial or extracranial pressure or as a result of cerebral edema and severe hypoxemia [2,3]. Transient MCA-REDF may occur due to excessive external probe pressure, which signifies an artifact without clinical consequence [4]. We reported a rare case of abnormal color Doppler sonography findings indicating a persistent REDF MCA in a 33-year-old female patient at 30 weeks gestation who was referred to our hospital with severe preeclampsia, fetal growth restriction, and oligohydramnios.

## 2. Case Presentation

Patient G2P1A0, age 33 years, with a gestational age of 30 weeks according to last menstrual period. She was referred to Hasan Sadikin Hospital because of high blood pressure. High blood pressure was detected in the seventh month of pregnancy. The presence of high blood pressure was accompanied by severe headaches, blurred vision, and heartburn. There was no complaint of labor contraction nor a history of profuse discharge from the birth canal. Fetal movement was still reported by the mother. A previous history of high blood pressure was denied; there was neither a history of high blood pressure in the family nor a history of other chronic diseases. The history of contact with COVID-19 patients was denied. The patient had a two-dose vaccination for COVID-19.

During the physical examination, the patient was fully alert, with a blood pressure of 180/110 mmHg, heart rate of 87 beats/min, respiration of 20 times/min, and a temperature of 36.6 °C. Obstetric examination revealed a uterine fundal height of 22 cm, an abdominal circumference of 97 cm, a fetal heart rate of 144–148 beats/min, and the longitudinal position of the fetus was 5/5 left back. The urine analysis was performed with proteinuria + 2.

Ultrasound examination showed a cephalic lie fetus with biometry (biparietal diameter (BPD), head circumference (HC), abdominal circumference (AC), and femur length (FL)) equal to 26–27 weeks of gestation; the gestational age based on the last menstrual period was 30 weeks. The Transcerebellar diameter was equal to 29–30 weeks of gestation, and the amniotic fluid volume suggested oligohydramnios with an amniotic fluid index (AFI) equal to 1.04 cm, with an estimated fetal weight (EFW) of 956 g (<10th percentile for gestational age) (Figure 1). The structure of both kidneys and urinary vesica was within a normal range, and fetal structural abnormalities were difficult to assess. The placenta was located anteriorly.

Color Doppler examination of the MCA revealed a peak systolic velocity of 42.33 cm/s, which is 1.045 times the median (the normal range is 40.5 cm/s for this gestational age), with persistent reversed end diastolic flow (Figure 2). The umbilical artery pulsatility index (PI) was 0.99, and RI was 0.63, with an S/D ratio of 2.71. There was notching on the right uterine artery, with a PI of 2.11. Notching was also found on the left uterine artery, with a PI of 1.58 (Figure 3). Laboratory results were within a normal range.

A cesarean section was performed due to severe preeclampsia with a low bishop score. Before the procedure, the patient’s condition was stabilized with the administration of a magnesium sulfate loading dose of 4 g IV over 15 to 20 min followed by a maintenance dose of 1 to 2 g/h. Methyldopa, as an antihypertensive drug, was administered to reduce the mean arterial pressure with a MAP reduction target of 25 percent. A male baby was born with a birth weight of 1100 g, and the baby’s length was 37 cm; the baby was in poor condition with an APGAR score = 1 in min 1 and min 5. Resuscitation was performed; however, the baby only survived for 10 min. The mother’s condition after delivery was stable, postoperative monitoring was carried out in the recovery room and continued care in the postpartum ward. The mother came home from the hospital in good condition.

## 3. Discussion

Currently, Doppler ultrasound is widely used to evaluate fetal circulation in both normal and pathological pregnancy. Fetal arterial Doppler studies are useful in determining the differential diagnosis of small-for-gestational-age fetuses. Due to impaired placental perfusion, the PI in the umbilical artery is increased and, in the fetal MCA, the PI is decreased. Hemodynamic changes occur in fetal arterial vessels during hypoxia and acidemia, induced by uteroplacental insufficiency [5,6].

MCA assessment is performed to assess fetal well-being at ultrasound. In healthy fetuses, there is a high resistance of cerebral blood flow with a continuous forward flow of the MCA. In a hypoxic fetus, cerebral resistance decreases while the peripheral resistance increases and the cerebral vasculature dilates, causing a reduction in the MCA pulsatility index in a phenomenon known as “brain sparing” [7,8,9]. With disease progression, the fetus is no longer able to compensate, and resistance in the MCA increases [8,10]. In the preterm (<32 weeks) small-for-gestational-age fetus, MCA Doppler has limited accuracy in predicting acidemia in the absence of other Doppler abnormalities. In the term small-for-gestational-age fetus with normal umbilical artery Doppler, an abnormal MCA Doppler (PI < 5th centile) has moderate predictive value for acidosis at birth and should be used to time delivery [4].

Reverse flow in the middle cerebral artery (MCA) is a rare condition found in fetuses. Some reports suggested severe neonatal morbidity or rapid intrauterine fetal death in association with fetal growth restriction or intrauterine pathology [2,3]. Small-for-gestational age fetuses may be constitutionally small, with no increased perinatal death or morbidity, or their growth may be restricted due to either low growth potential, the result of genetic disease or environmental damage, or due to reduced placental perfusion and uteroplacental insufficiency. Preeclampsia and fetal growth restriction are associated with inadequate quality and quantity of the maternal vascular response to placentation. In both conditions, there are characteristics of pathological findings in the placental bed. In preeclampsia, there is a necrotizing lesion with foam cells in the wall of the basal and spiral arteries, which is referred to as acute atherosclerosis. In essential hypertension, there are hyperplastic lesions in the basal and spiral arteries [5].

The exact pathophysiology of reverse flow in the MCA is still not fully understood. Failure of cerebral circulation can be caused by extracranial pressure such as external artificial pressure, oligohydramnios, and anhydramnios, or increased intracranial pressure due to hydrocephalus, cerebral edema, or cerebral hemorrhage [3,6,7]. Transient REDF MCA can occur due to an artifact produced by excessive external pressure by the probe, which has no impact or clinical consequences. If such an artifact occurs, the examination should be cautiously repeated and confirmed by the second examiner to minimize the pressure by the transducer. Persistent REDF MCA is diagnosed if the examination is conducted properly with an appropriate technique and following the description of persistent REDF MCA [7,8].

REDF in the MCA is an advanced hemodynamic disorder. Mari and Wassertrum were the first to report the loss of brain sparing before death in fetuses with fetal growth restriction conditions, and the observation results were also confirmed by several other examiners. These findings signify increased intracranial pressure due to cerebral edema associated with severe hypoxemia, as indicated by Vyas et al., who demonstrated the correlation of PI in the MCA and blood gas level from fetal blood sampling. In mild-to-moderate hypoxemia, there is a decrease in the PI of MCA with cerebral vasodilation. However, in the prolonged hypoxemia condition, the brain-sparing effect autoregulation mechanism cannot sufficiently compensate due to increased intracranial pressure following cerebral edema. Thus, the PI of MCI decreases in fetuses with highly severe hypoxemia [8].

Hypoxemia is compensated by a central redistribution of blood flow, resulting in increased blood flow to the brain, heart, and adrenals, and decreased blood flow to the peripherals [1,5]. This mechanism allows preferential delivery of nutrients and oxygen to vital organs. Increased blood flow of MCA in Doppler findings is characterized by a decrease in PI with manifestations of cerebral vasodilation as a response to fetal hypoxemia [2,9,10]. Compensation through cerebral vasodilation is limited, and a plateau corresponding to the base of PI level is reached at least 2 weeks before the development of the fetus is compromised [5]. In advanced cases, this autoregulation process does not adequately perform, and there is an increase in resistance to cerebral blood flow, leading to a poor fetal state. Reverse flow in MCA is a rare case indicating the loss of brain-sparing autoregulation, which is strongly associated with the impending death of the fetus [11].

There have been several cases of PREDF in MCA (Table 1). The diagnosis mainly occurs in the third trimester. The etiologies include intracranial hemorrhage, severe anemia due to feto-maternal hemorrhage or severe fetal growth restriction, and a rare intrahepatic bile duct malformation. Obeidat et al. reported a case with hydrocephalus as the etiology [12]. All cases lead to severe outcomes for the fetus or neonate, such as intrauterine death or morbidity. Fetal morbidities include a grade III intraventricular hemorrhage, periventricular leukomalacia, hemorrhagic parietal infarct, and bilateral ischemic changes in the basal ganglia.

Under conditions of prolonged severe hypoxemia, cerebral oxygen consumption decreases, while anaerobic glycolysis increases, leading to swelling and cell death. Cerebral vasodilatation ceases because of the increase in intracranial pressure resulting from cerebral edema. If the cerebral resistance continuously rises, a loss of autoregulation of the cerebral blood flow occurs, and reverse flow begins. Fetomaternal hemorrhage causes fetal anemia that precipitates cerebral ischemia or hemorrhage, or both, resulting in cerebral edema [11]. The increased intracranial pressure results in the reversed end-diastolic flow of the MCA. The bleeding volume from intracranial hemorrhage causes increased intracranial pressure. In congenital hydrocephalus, excessive accumulation of cerebrospinal fluid also causes increased intracranial pressure [11].

Sevulpeda et al. previously reported PREDF in MCA caused by severe fetal growth restriction with intrauterine fetal death outcomes; meanwhile, Kawakita et al. reported PREDF in MCA, but without recorded neonatal follow-up outcomes [4,8]. In this case, a pregnancy with severe preeclampsia, fetal growth restriction, and oligohydramnios has Doppler findings indicating PREDF, resulting in early neonatal death (within 10 min). This is in line with previous case reports where PREDF in MCA in fetuses with fetal growth restriction have very poor outcomes, leading to intrauterine/early neonatal death.

The Doppler examination also revealed a notching of the right and left uterine arteries, which is a marker of spiral artery remodeling failure, causing early-onset severe preeclampsia [11,12]. In this case, inadequate early prevention worsened the patient’s condition, leading to severe preeclampsia. The pregnancy did not present with hydrocephalus, cerebral hemorrhage, or mechanical compression of the fetal head. Another anatomical abnormality cannot be assessed intrauterine, nor seen after delivery. We did not perform a postnatal examination to exclude any internal neonatal anatomy abnormalities.

## 4. Conclusions

The findings of PREDF, in this case, can be caused initially by spiral artery remodeling failure that leads to severe preeclampsia and early onset fetal growth restriction, then further makes microvasculature disturbance in the renal artery that causes oligohydramnios. All of these conditions lead to severe hypoxemia, cerebral edema, and eventually increased intracranial pressure. PREDF in MCA in this pregnancy indicated an advanced stage of vascular disturbance with a poor outcome. Operative delivery was performed in favor of ameliorating the mother’s condition, presenting with severe preeclampsia and a low bishop score after the stabilization phase. Further research is necessary to outline various causes precipitating PREDF in MCA and its association with fetal/neonatal conditions so we can formulate the most suitable treatment for this condition.

## Figures and Tables

**Figure 1 medicina-59-01648-f001:**
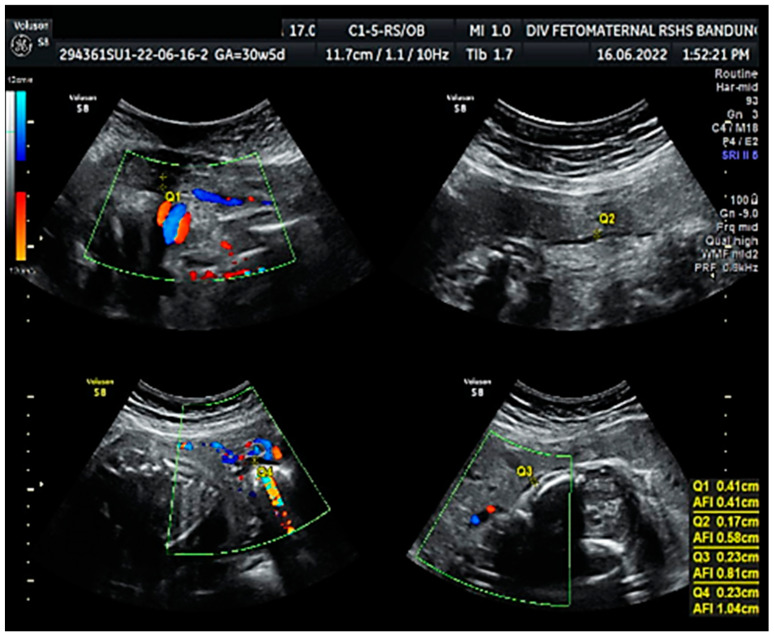
Amniotic fluid index (AFI 1.04 cm): oligohydramnios.

**Figure 2 medicina-59-01648-f002:**
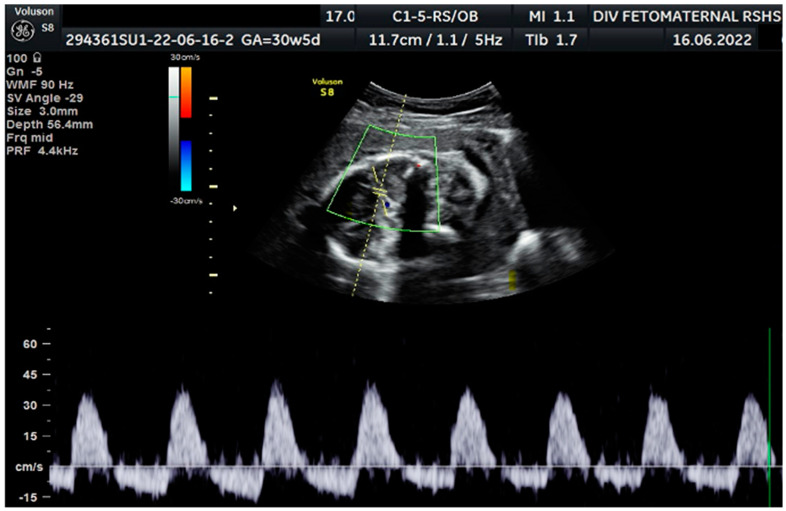
Reversed end diastolic flow in the middle cerebral artery.

**Figure 3 medicina-59-01648-f003:**
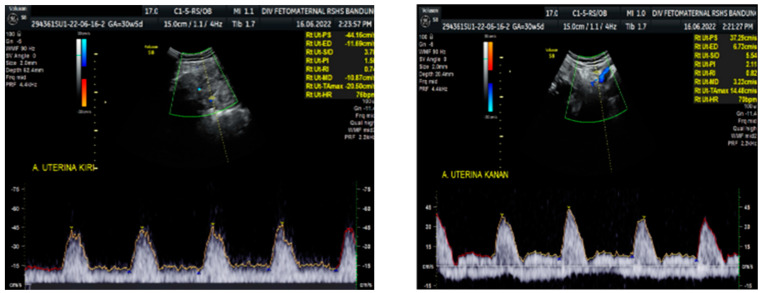
Notching on right and left uterine artery Doppler.

**Table 1 medicina-59-01648-t001:** Case reports on PREDF in the MCA and its outcome.

No	Article	Gestation	Management	Cause	Outcome
1	Hirshberg et al. [2]	27 + 5 weeks gestation	Expectative	Intraventricular hemorrhage	In-utero fetal death
2	Browfoot et al. [4]	32 weeks gestation	Cesarean section	Feto-maternal hemorrhage	Hemorrhagic parietal infarct and bilateral ischemic changes in the basal ganglia
3	Respondek et al. [7]	31 weeks gestation	Expectative	Rhesus disease	In-utero fetal death
4	Sepulveda et al. [8]	30 weeks gestation	Expectative	Severe fetal growth restriction	In-utero fetal death
5	Giancotti et al. [9]	27 weeks gestation	Expectative	Intrahepatic bile duct malformation	Neonatal death
6	Chainarong et al. [11]	33 weeks gestation	Cesarean section	Intracranial hemorrhage	Neonatal death
7	Kawakita et al. [13]	27 + 3 weeks gestation	Cesarean section	Severe fetal growth restriction	No neonatal follow-up up recorded
8	Baschat et al. [14]	29 + 2 weeks gestation	Cesarean section	Feto-maternal hemorrhage	Intraventricular hemorrhage, periventricular leukomalacia, moderate respiratory distress syndrome
9	Our case	30 weeks gestation	Cesarean section	Severe fetal growth restriction	Neonatal death

## Data Availability

Data availability will be available on the hospital of the study. Contact the author for future access.

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
