# Peer review of "Persistent Reverse End Diastolic Flow in Fetal Middle Cerebral Artery: A Rare Finding with Poor Outcomes"

_medicina, 2023, doi:10.3390/medicina59091648_

Round 1

Reviewer 1 Report

Kindly revise the manuscript for grammatical and language mistakes.

The punctuation marks are not adequately put. Please consider reviewing the manuscript for punctuation flaws.

Author Response

Thank you for the input, we have been adding and adopting it.

Reviewer 2 Report

With pleasure, I read the paper titled: “PERSISTENT REVERSE END DIASTOLIC FLOW IN FETAL MIDDLE CEREBRAL ARTERY: A RARE FINDING WITH POOR OUTCOMES”. Overall, the paper reads well. The major strength lies in being among the first few studies report PREDF. Collectively, the article is well-written with good flow of ideas, proper English language, up-to-date citations, and focused figures. I have the following comments:

1. Introduction. Please clearly indicate the significance of the study by highlighting how many previous cases have been reported previously? Also, please briefly summarize the findings of previous investigations and emphasize your case report is different.

2. Case report. Please provide more information on the status of the mother after delivery.

3. Discussion. For table 1, you cited Brownfoot et al and summarized his findings. I believe you should expand on the Table by summarizing additional case reports and adding more details such as gestational age, pregnancy conditions, possible etiology, interval between diagnosis and adverse event, and etc. There are more case reports published that should be acknowledged and summarized (doi: 0.1016/j.radcr.2023.02.028; doi: 10.1016/j.ejogrb.2013.07.046, doi: 10.1002/jcu.23112, and doi: 10.1046/j.1469-0705.2000.00167.x). Then it is important to pinpoint the similarities and differences with the existing literature to highlight the unique aspects of your case report.

 Minor editing of English language required

Author Response

(The authors gave the same response as above.)

Reviewer 3 Report

The authors report on a growth restricted fetus, induced by severe preeclampsia, revealing a persistent reversed end diastolic flow (PREDF) in middle cerebral artery (MCA). This aspect is, indeed, a rare and interesting finding.

 Comments 

1. Although PREDF in MCA is a very serious condition, I consider that the statement: “(PREDF) … is the terminal hemodynamic event” (line 14), is too strong and the phrase could be nuanced.

 2. The authors should report data about the actual pregnancy (before the present admission – lines 47-55).

 3. All along the manuscript, the authors report on the IUGR, induced by the severe preeclampsia. Furthermore, they report that, at admission, the fetal ultrasound evaluation revealed: “Biparietal Diameter (BPD), Head Circumference (HC), Abdominal Circumference (AC), and Femur Length (FL) corresponding to 26-27 weeks of gestation; the gestational age based on the last menstrual period was 30 weeks” (lines 62-65).

3.1. It is unclear if BPD, HC, AC and FL were all concordant with 26-27weeks.

3.2. Furthermore, regarding the same comment, the authors reported that: “transcerebellum diameter corresponded to 29-30 weeks of gestation” (lines 65-66).

The authors should detail the fetal biometry.

 2. If authors evaluated that PREDF in MCA is a “terminal hemodynamic event”, it is perfectly logic that the rapid delivery was decided and, due to the poor maternal state, the C-section was the right decision.

2.1. The authors have to mention this aspect and to precisely mention the interval from admission to delivery.

2.2. Being a very serious condition, I assume that at least one expert in fetal-maternal ultrasound made the fetal evaluation and monitoring, in such serious condition: severe preeclampsia with (severe) IUGR. The authors should explain the significant difference, between the ultrasound estimated fetal weight and the newborn’s weight: 956g vs. 1.100g.

2.3. May be, the ultrasound scanner report on the percentile, regarding the fetal weight, would be interesting for the reader.

 3. The authors should detail about the “given antihypertensive drugs”.

Author Response

(The authors gave the same response as above.)

Round 2

Reviewer 2 Report

The authors did a great job by addressing the raised concerns by the pee-reviewers and substantially enhanced the overall readout of the article. The manuscript is now more scientifically sound and solid, and likely to be cited in the future. Well-done!

Minor to moderate English editing is required